# Association of the Omega-3 Index with Incident Prostate Cancer with Updated Meta-Analysis: The Cooper Center Longitudinal Study

**DOI:** 10.3390/nu13020384

**Published:** 2021-01-26

**Authors:** Stephen W. Farrell, Laura F. DeFina, Nathan L. Tintle, David Leonard, Kenneth H. Cooper, Carolyn E. Barlow, William L. Haskell, Andjelka Pavlovic, William S. Harris

**Affiliations:** 1The Cooper Institute, Dallas, TX 75230, USA; sfarrell@cooperinst.org (S.W.F.); ldefina@cooperinst.org (L.F.D.); dleonard@cooperinst.org (D.L.); bwright@cooperinst.org (C.E.B.); apavlovic@cooperinst.org (A.P.); 2Fatty Acid Research Institute, Sioux Falls, SD 57106, USA; nlt@faresinst.com; 3Department of Mathematics & Statistics, Dordt University, Sioux Center, IA 51250, USA; 4Cooper Clinic, Dallas, TX 75230, USA; khcooper@cooper-clinic.com; 5Department of Medicine, Stanford University, Palo Alto, CA 94305, USA; whaskell@stanford.edu; 6Department of Internal Medicine, Sanford School of Medicine, University of South Dakota, Sioux Falls, SD 57105, USA

**Keywords:** prostate cancer, omega-3 fatty acids, eicosapentaenoic acid, docosahexaenoic acid, epidemiology, biomarker, prospective cohort

## Abstract

Background: The association between long-chain omega-3 polyunsaturated fatty acids (n-3 PUFA) and prostate cancer (PC) remains unclear. Methods: We compared incident PC rates as a function of the Omega-3 Index [O3I, erythrocyte eicosapentaenoic and docosahexaenoic acids (EPA + DHA)] in 5607 men (40–80 years of age) seen at the Cooper Clinic who were free of PC at baseline. The average follow-up was 5.1 ± 2.8 years until censoring or reporting a new PC diagnosis. Proportional hazards regression was used to model the linear association between baseline O3I and the age-adjusted time to diagnosis. A meta-analysis of n-3 PUFA biomarker-based studies and incident PC was updated with the present findings. Results: A total of 116 cases of incident PC were identified. When O3I was examined as a continuous variable, the age-adjusted hazard ratio (HR) (95% CI) was 0.98 (0.89, 1.07; *p* = 0.25) for each 1% increment in the O3I. The updated meta-analysis with 10 biomarker-based studies found no significant relationship between EPA or DHA levels and risk for PC. Conclusions: We find no evidence in this study nor in a meta-analysis of similar studies that consuming n-3 PUFA-rich fish or using fish oil supplements affects the risk of PC.

## 1. Introduction

Prostate cancer (PC) is the second leading cause of cancer death in U.S. men, with annual mortality of approximately 32,000 [1]. Nearly 175,000 new cases of PC are diagnosed in the U.S. each year. As inflammation plays a role in the etiology of many types of cancer, several PC-related studies have focused on factors associated with inflammation. These include lifestyle-related variables such as diet [2] and obesity [3], as well as medications such as statins [4]. While obesity has been shown to be associated with inflammation and greater risk of high-grade PC [5] and PC death [6], the association of these other variables, particularly diet, with PC is inconsistent.

From a dietary standpoint, there has been considerable interest in the association of long-chain omega-3 polyunsaturated fatty acids (n-3 PUFA) with PC incidence. These fatty acids are found primarily in cold water fatty fish and dietary supplements and have been shown to have anti-inflammatory effects through their impact on prostaglandins and other oxylipins [7].

In 2013, Brasky and colleagues [8] published a case-cohort study in which men in the highest quartile of plasma n-3 PUFA levels were found to have an increased risk of low grade, high grade, and total PC when compared to men in the lowest quartile. These authors went on to state that their findings suggested that “these fatty acids are involved in prostate tumorigenesis,” and further, that recommendations “to increase long chain n-3 PUFA intake should consider its potential risks.” Although this report was widely criticized (i.e., there was no measure of n-3 PUFA intakes, causation was inferred from an association, and case-control blood levels differed by only 2% [9,10,11]), it nevertheless attracted considerable attention, particularly in the lay press, and cast a serious pall over the prevailing view that the n-3 PUFAs were generally healthy. Subsequent meta-analyses have found small and non-significant direct relationships between circulating n-3 PUFA levels and total PC risk [12,13]. Hence, the question remains open and further studies are needed. Studies reporting on the association between dietary n-3 PUFA with PC incidence are inconsistent [8,14] perhaps due in part to the questionable validity of questionnaires that are utilized to estimate dietary intake [3]. A clearer picture of the relationship between n-3 PUFAs and disease may be obtained using an objective measure of blood n-3 PUFA levels. In 2004, Harris and von Schacky proposed the Omega-3 Index (O3I) as a measure of n-3 PUFA status [15]. The O3I is the percentage of fatty acids in erythrocyte membranes that are composed of eicosapentaenoic acid (EPA) and docosahexaenoic acid (DHA). In 2016, our group reported no significant cross-sectional associations between erythrocyte n-3 PUFA, including the O3I, and prostate specific antigen (PSA) levels in a sample of 6219 men without PC [16]. However, because the literature regarding n-3 PUFA and incident PC remains inconsistent, the purpose of the current investigation is to examine the association between baseline O3I and incident PC in a sample of healthy men who presented for a comprehensive medical examination.

## 2. Materials and Methods

The Cooper Center Longitudinal Study (CCLS) is a prospective epidemiological study of health characteristics, health behaviors, and chronic disease biomarkers in adult men and women. Participants in the CCLS are generally healthy individuals who are either self-referred or referred by their employers for preventive medical examinations. All procedures related to the CCLS are reviewed and approved by The Cooper Institute’s institutional review board on an annual basis. After receiving written informed consent from each participant, a clinical evaluation was performed, which included an examination by a physician, fasting blood chemistry assessment, personal and family health history, anthropometry, resting blood pressure, electrocardiogram, and a maximal graded treadmill exercise test. PSA measurement was performed using the chemiluminescent method on an Abbott Architect CI8200 Analyzer.

Prior to the treadmill exercise test, and following a 10–12 h fast, blood was drawn into an EDTA tube. Tubes were sent overnight on ice to OmegaQuant Analytics, LLC (Sioux Falls, SD, USA) for analysis of the O3I. After isolation of the erythrocytes by centrifugation, the cells were analyzed for fatty acid composition as described previously [17]. The coefficient of variation for this assay is 2.4%. Two different O3I categorization strategies were utilized for the present study. The first categorization used O3I values of <4%, 4–8%, and >8% [18], while the second used quintiles of O3I. A continuous version of the O3I was utilized as well. Incident PC was identified during the follow-up visit via self-report utilizing a medical history questionnaire.

Participants in the present study were selected from 6274 men between 40 and 80 years who had at least two visits to the Cooper Clinic spanning a period of at least one year between 2007 and 2019. Patients with prevalent CVD (*n* = 76), history of PC (*n* = 224), those with body mass index values < 18.5 kg/m^2^ (*n* = 122), missing PSA values (*n* = 51), or with PSA values >4.0 ng/mL (*n* = 194) were excluded from the analyses. Incident cases were defined as men reporting no PC at the initial visit and a new diagnosis of PC at a follow-up visit. These exclusions resulted in a final sample of 5607 men.

### Statistical Methods

Age at time of PC diagnosis was predicted by categories of the O3I, by quintiles and by continuous O3I (per 1% increment) using proportional hazards models with interval censoring. Timing of PC diagnoses after the baseline visit was based on the year of diagnosis self-reported at follow-up visits. The date of diagnosis was therefore narrowed to a censoring interval of at most 1 year among the cases. When a new PC diagnosis was reported without specifying the year of diagnosis, the censoring interval extended to the previous clinic visit. For those reporting no PC during follow-up, PC-free survival was right-censored at the date of their last follow-up visit. Adjusted models for the predicted age analysis included BMI and baseline PSA. We also conducted time-to-diagnosis analyses in unadjusted, age-adjusted and age, BMI and PSA-adjusted models. Alcohol use and smoking were considered as potential additional model covariates, but according to the American Cancer Society [19] and to sensitivity analyses in a recent meta-analysis of biomarker-based observational studies [13], there is little evidence that these are important risk factors for PC. We nevertheless sought to confirm this in the current study by comparing these metrics (by t-test) in those reporting and those not reporting incident PC. A significance level of 0.05 was used for all analyses. The ICsurv package in R was used for proportional hazards regression modeling [20].

We also conducted a meta-analysis of PC risk by incorporating the CCLS results, presented here for the first time, into a recently published meta-analysis [7,13] that included nine previously published studies (a mix of prospective cohort studies and case-control studies). Following Fu et al. [13], models were fit predicting risk for PC with separate models for DHA and EPA. The 10 cohort-specific hazard ratios (HRs) were pooled by inverse-variance weighted meta-analysis. Heterogeneity was assessed by the I^2^ statistic and Q-test. Again, a significance level of 0.05 was used for the meta-analysis. The Metafor package in R was used for all meta-analyses [20].

## 3. Results

Characteristics of the sample are presented in Table 1. The mean age was 53 ± 8 years, and 93% were White. A total of 116 cases of incident PC were identified during an average follow-up time of 5.1 ± 2.8 years. Neither alcohol intake nor smoking differed significantly between those who did and did not develop PC. For alcoholic drinks per week, values were 7.1 ± 6.2 and 6.7 ± 6.7 (*p* = 0.6), respectively, and rates of smoking were 6.9% and 9.6% (*p* = 0.32), respectively. The unadjusted O3I was higher in the 116 cases than in the non-cases (by 0.35%, *p* = 0.048), but after adjusting for age, there was no difference (0.004%, *p* = 0.98).

As shown in Table 2, whether analyzed by category of the O3I, by quintile of O3I or as a continuous variable, the O3I was not significantly associated with a PC diagnosis. This finding was true regardless of whether the models predicted age at diagnosis or time to diagnosis and regardless of covariate adjustment. When age was not included in the time to diagnosis model, risk appeared to trend directly with the O3I, however, after age adjustment, the trend reversed. With further adjustment for BMI and PSA levels, HRs tended to become slightly lower. Indeed, in all analyses that included age (O3I by category, by 1% increment, and by quintile), the trends, although not statistically significant, favored lower risk with higher O3I levels.

In the current updated meta-analysis, data were only available to compare risk for PC on the basis of a per-1% increment in DHA and in EPA, not the O3I. Nevertheless, including the data from the present study, there were no statistically significant relationships between risk for PC and blood levels of either EPA (*p* = 0.20) or DHA (*p* = 0.11) (Figure 1).

## 4. Discussion

Although inflammation plays a role in the etiology of PC [21], and n-3 PUFA are associated with reduced inflammation [7], the literature is inconsistent with regard to the relationship between n-3 PUFA and incident PC. Due to this inconsistency and the ambiguous findings in past meta-analyses [13,14], we examined this relationship in a group of 5607 apparently healthy men. In our study, we found no significant association between baseline O3I and incident PC over a mean follow-up period of 5.1 years. The HRs presented in Table 2 (although uniformly non-significant) require some interpretation. For example, the HR of 0.95 in the age of diagnosis model treating the O3I as a continuous exposure variable means that between two men with a 1% difference in the O3I and otherwise equivalent, the man with the higher O3I was 5% less likely to develop incident PC. In the categorical analysis, where the HR was 0.65 for an O3I of >8%, this indicated that a man in that O3I category has a 35% lower risk of incident PC than a man with an O3I of <4%. Indeed, after adjusting for age, there was no difference in the O3I between cases and controls.

Earlier work in this area has been summarized in meta-analyses of 16 omega-3/fish intake-based studies [14] and nine circulating biomarker-based studies [13]. In the former, the authors concluded that there was insufficient evidence to conclude that a relationship existed between omega-3 fatty acid intake and PC; in the latter, there was a marginal (i.e., 2%) increased risk for PC per 1% increase in plasma DHA levels but no increased risk associated with EPA levels. Interestingly, circulating omega-3 levels (whether in plasma or erythrocyte membranes) may not reflect fatty acid levels in the prostate gland itself. Moussa et al. reported in the study of men with low-risk PC that higher prostate levels of EPA were associated with lower odds of progressing to high-risk PC, but erythrocyte EPA levels were unrelated with risk [22]. Hence, the fundamental assumption of blood-derived, biomarker-based studies may need to be reconsidered in PC research.

Among the biomarker-based studies, that of Brasky and colleagues [8] noted earlier raised significant concern in both the medical and lay press regarding recommendations to increase n-3 PUFA intakes for the prevention of cardiovascular disease. In that case-cohort study, the estimated O3I in the men without PC was 4.4% and that for the men with PC was 4.5%. Nevertheless, in their statistical models, men in the highest quartile of plasma phospholipid n-3 PUFA levels had a HR of 1.43 (1.09–1.88) for PC when compared to the men in the lowest quartile. In their discussion, the authors concluded that the dietary intake of omega-3 (fish and/or supplements) could potentially cause prostate cancer. Even if such speculation was true, and the slightly higher n-3 PUFA levels were caused by a higher intake of farmed salmon, it would be difficult to separate the effects of the n-3 PUFAs provided by the salmon from the effects of multiple environmental pollutants [23] that may have been common in farmed salmon in those days [24]. In addition, there are other possible explanations for their observation. For example, there are reports [25,26] that in precancerous tissues, activity of delta-6-desaturase is increased. This enzyme is rate-limiting for the synthesis of long-chain n-6 and n-3 PUFAs from linoleic acid and alpha-linolenic acid, respectively. Hence, in the metabolic milieu of cancer cells, an increase in tissue (and possibly plasma) levels of EPA and DHA could be an effect, not a cause, of the relationship reported by Brasky et al. Regardless, the findings of the current study do not support the observations of that study.

We added the present findings to the most recent meta-analysis of studies examining the associations of circulating levels of n-3 PUFAs and risk of PC [13]. With now 10 studies contributing, we found no significant association of n-3 PUFA status and risk of incident PC.

Strengths of the current study include a large sample size with an objective baseline measure of n-3 PUFA status in lieu of dietary or supplement use self-report, a focus on generally healthy men, and the inclusion of an updated meta-analysis of studies examining this question. There are also several limitations to the current study. The sample was primarily White with a relatively high socioeconomic status (SES), which limits our external validity. We cannot state for certain that these findings would be observed in other ethnic groups, or in groups with lower SES. However, this same limitation strengthens our internal validity. Although digital rectal exams (DREs) were included as part of the baseline medical exam in this sample, DRE results have not been entered into the CCLS database until very recently. Thus, it is possible that men with PSA values < 4 ng/mL but with an abnormal DRE were misidentified as being free from PC at baseline. PC incidence was self-reported using a medical history questionnaire at follow-up. It is possible that some men may have incorrectly reported a diagnosis of PC during follow-up. However, the likelihood of this occurrence is very low due in part to the high SES (for example, the mean education level was 16.8 ± 2.0 years) of the sample. Furthermore, mean PSA values were significantly higher at the second visit for men with incident PC (1.7 ng/mL) versus men who were cancer free at the second visit (1.0 ng/mL, *p* < 0.001). Additionally [27], we have shown that Cooper Clinic patients demonstrated a sensitivity of 98% when self-reporting incident hypertension. Another limitation is that the timing of PC diagnosis was known only to within 1 year. The five-year follow-up time was relatively short for men of average age 50, especially since PC is more likely to develop in older men. Finally, the self-reported nature of PC incidence did not include information regarding stage or grade, i.e., Gleason Score.

## 5. Conclusions

In conclusion, we found no significant association between objectively measured n-3 PUFA status and incident PC in a group of apparently healthy men. Furthermore, when the results of this study were pooled with prior literature, no significant relationship between blood levels of omega-3 fatty acids and PC was found. Therefore, concerns about increasing the risk of PC should not dissuade health care professionals from continuing to recommend 1–2 servings per week of non-fried fatty fish to their male patients for cardiovascular benefit per the 2018 advisory from the American Heart Association [28] and from the 2015–2020 Dietary Guidelines for Americans [29].

## Figures and Tables

**Figure 1 nutrients-13-00384-f001:**
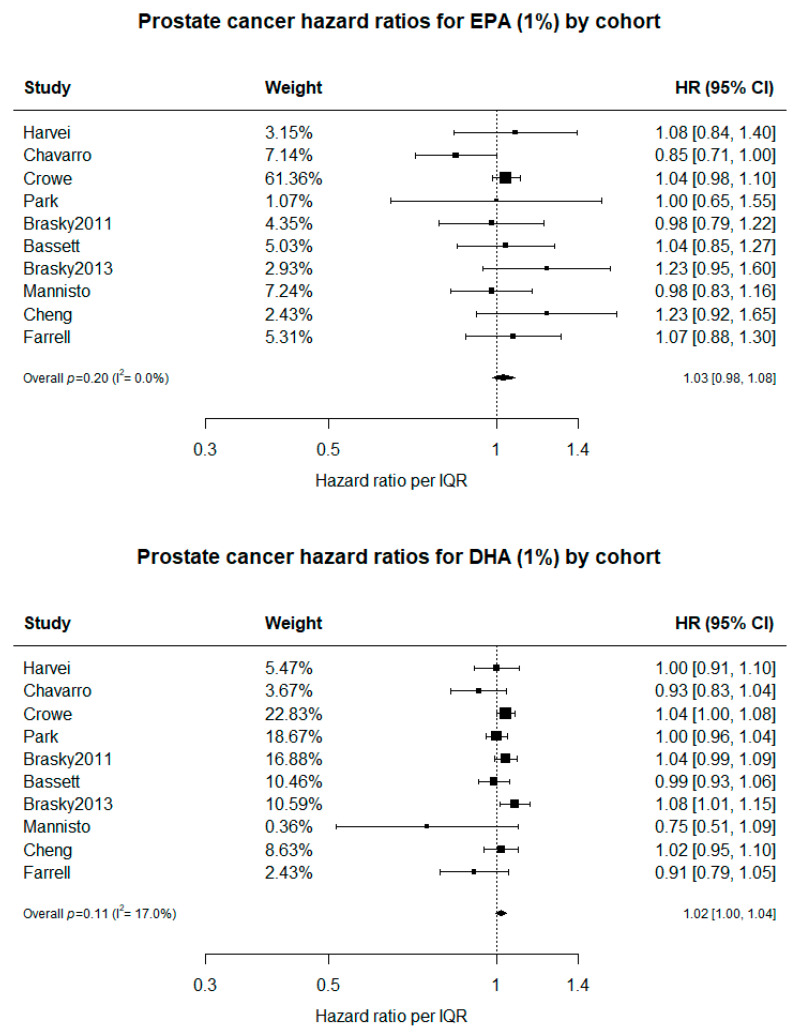
Hazard ratios with 95% CIs are presented by cohort to predict the risk of PC by EPA (Figure 1a) and, separately, DHA (Figure 1b). Neither EPA (*p* = 0.20) nor DHA (*p* = 0.11) shows evidence of association with PC risk, with little evidence of study-to-study heterogeneity in risk estimates. All data (except Farrell which are the present results) were taken from Fu et al. [13] where citations to the original studies may be found.

**Table 1 nutrients-13-00384-t001:** Baseline characteristics of 5607 men. The Cooper Center Longitudinal Study (CCLS), 2007–2019 ^1^.

Variable	Value
Age (years)	53.4 (8.4)
White (332 missing) (%)	4929 (93.4%)
Omega-3 Index (%)	6.0 (1.9)
BMI (kg/m^2^)	27.7 (3.9)
Maximal METS (674 missing)	11.5 (2.1)
Glucose (mmol/L)	5.4 (0.8)
Total cholesterol (mmol/L)	4.8 (1.0)
HDL cholesterol (mmol/L)	1.4 (0.4)
Triglycerides (mmol/L)	1.3 (0.9)
Systolic BP (mmHg)	120.8 (12.0)
Diastolic BP (mmHg)	79.8 (8.7)
Prostate-specific antigen (ng/mL)	1.1 (0.7)
Alcohol (drinks/week)	6.7 (6.7)
Current Smoking, *n* (%)	537 (9.6%)
Omega-3 supplement use (369 missing), *n* (%)	2107 (40.2%)

^1^ Data are presented as means (SD) or *n* (%).

**Table 2 nutrients-13-00384-t002:** Risk for a prostate cancer diagnosis and the Omega-3 Index (O3I): hazard ratios (HRs) from models for age of diagnosis and time to diagnosis by quintile, by category and continuously (per unit (1%) increment in O3I).

		Modeling Age of Diagnosis		Modeling Time to Diagnosis		
Omega-3 Index		HR (95% CI)	Adjusted HR ^1^ (95% CI)	Unadjusted HR (95% CI)	Age-Adjusted HR (95% CI)	Adjusted HR ^2^ (95% CI)
Quintile	Q1: O3I < 4.4	1	1	1	1	1
	Q2: 4.4 ≤ O3I <5.3	0.54 (0.27,1.07)	0.52 (0.26,1.04)	0.64 (0.32,1.28)	0.57 (0.29,1.14)	0.54 (0.27,1.08)
	Q3: 5.3 ≤ O3I <6.3	0.72 (0.39,1.30)	0.72 (0.39,1.31)	0.94 (0.52,1.71)	0.78 (0.43,1.42)	0.79 (0.43,1.44)
	Q4: 6.3 ≤ O3I <7.6	0.67 (0.37,1.21)	0.64 (0.36,1.16)	0.99 (0.55,1.78)	0.75 (0.42,1.36)	0.71 (0.39,1.29)
	Q5: 7.6 ≥ O3I	0.61 (0.34,1.07)	0.55 (0.31,0.98)	1.17 (0.67,2.07)	0.73 (0.41,1.30)	0.65 (0.36,1.16)
	P for trend	0.2	0.13	0.25	0.6	0.36
Category	O3I < 4%	1	1	1	1	1
	O3I 4–8%	0.73 (0.41,1.30)	0.70 (0.39,1.26)	1.02 (0.57,1.83)	0.81 (0.45,1.46)	0.78 (0.44,1.41)
	O3I > 8%	0.65 (0.33,1.27)	0.60 (0.30,1.19)	1.31 (0.67,2.58)	0.79 (0.4,1.56)	0.72 (0.36,1.43)
	P for trend	0.2	0.17	0.2	0.23	0.38
Continuous	(per 1%)	0.95 (0.86,1.04)	0.93 (0.85,1.03)	1.06 (0.97,1.17)	0.98 (0.89,1.07)	0.96 (0.87,1.06)

^1^ Adjusted for BMI and prostate specific antigen (PSA). ^2^ Adjusted for Age, BMI and PSA.

## Data Availability

The data used for the analyses in this study may be available to other researchers upon receipt and approval of an application by the Cooper Institute.

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
