# Peer review of "Association of the Omega-3 Index with Incident Prostate Cancer with Updated Meta-Analysis: The Cooper Center Longitudinal Study"

_nutrients, 2021, doi:10.3390/nu13020384_

Round 1
Reviewer 1 Report
General comment:
This is an interesting and well-written manuscript, which assesses the association of the omega-3 index (O3I) with the incidence of prostate cancer (PCa) in the Cooper Longitudinal Study (CCLS) using also meta-analysis. Contrary to other studies, in particular that of Brasky et al. (2013), the results do not show a significant relationship between a high O3I and PCa risk. However, this study has some design weaknesses and raises some methodological questions.
Detailed comments:
Abstract
Line 28: Wouldn't it be better to say "affect" rather than “increase” since there was no significant results ?
Introduction
Line 39: It is said that the other variables, i.e. diet, obesity and statins, are inconsistent. However, the article that the authors quote shows that “statins reduce the risk of both total PC and clinically important advanced PC.” Please, explain why you cited this article on statins in the context of the association of O3I with PCa incidence.
Material and Methods
Line 77: Wouldn't it have been more judicious to repeat the O3I analysis on the second visit to confirm that the values obtained during the first O3I analysis did indeed reflect a different dietary habit between individuals with or without a PCa diagnosis?
Line 77: It is not said how the fact that the blood samples were taken before a treadmill exercise and after a 10-12 hour fast may affect the O3I analysis?!
Line 80: There is no reference after "as described previously"?! Does OmegaQuant Analytics LLC use the same gas chromatographic method as described in reference 17? If so, it would be useful to mention it more clearly in the “Materials and Methods” section.
Lines 87-88: What is the rationale for excluding patients with prevalent CVD and BMI <18.5?
Discussion
The authors suggest that the difference observed between the study by Brasky et al. (2013) can be explained by the fact that the high levels of EPA and DHA in tissues and plasma may be an effect and not a cause of PCa due to increased delta-6-desaturase activity. It is therefore a pity that they did not answer this question by performing another analysis on the second visit.
What about industrial chemicals or pollutants found in high levels in farm raised fish, like PCBs, which were described to be associated with PCa (Ali I et al. Carcinogenesis, 2016, 37(12): 1144–1151)? Could this potential factor also affect the results of Brasky et al.?
The authors describe their population as being men of high social class and undoubtedly concerned by their health, of which 40.2% use omega-3 supplements. Maybe these people are not consuming too much farmed raised fish at affordable price but containing carcinogenic molecules, compared to other populations ?!
Line 198: The parallel between self-reporting incident hypertension and PCa is not very relevant and seems to be only intended for self-citation.
Author Response
Newly added, not requested:
New line 119 (Results): The unadjusted O3I was higher in the 116 cases than in the non-cases (by 0.35%, p=0.048), but after adjusting for age, there was no difference (0.004%, p=0.98).
Reviewer 1:
Line 28: Wouldn't it be better to say "affect" rather than “increase” since there was no significant result?
Yes, that is reasonable and the change has been made.
Line 39: It is said that the other variables, i.e. diet, obesity and statins, are inconsistent. However, the article that the authors quote shows that “statins reduce the risk of both total PC and clinically important advanced PC.” Please, explain why you cited this article on statins in the context of the association of O3I with PCa incidence.
Topic here in the Introduction is that one of the mechanisms causing PC is inflammation, and so we wanted to support that by including a citation that drugs with anti-inflammatory effects (like statins) have been shown to help lower risk. But to your point, the citation we used was not the most relevant, and so we have substituted a newer reference that more directly supports our claim: K. Koushki, et al. Anti-inflammatory Action of Statins in Cardiovascular Disease: the Role of Inflammasome and Toll-Like Receptor Pathways, Clinical reviews in allergy & immunology, (2020).
Line 77: Wouldn't it have been more judicious to repeat the O3I analysis on the second visit to confirm that the values obtained during the first O3I analysis did indeed reflect a different dietary habit between individuals with or without a PCa diagnosis?
Agreed, a follow-up test would have been helpful to confirm the stability of the O3I within persons over time and to be surer that the baseline measure reflected steady state. Unfortunately, such data are not available.
Line 77: It is not said how the fact that the blood samples were taken before a treadmill exercise and after a 10-12 hour fast may affect the O3I analysis.
It’s unclear why anything needs to be said about it because the O3I was assessed in fasting samples, drawn before the treadmill test. This is the normal way it is analyzed. It is standard procedure in the Cooper Clinic for all bloodwork (lipids, glucose, O3I, etc) for the exam to be drawn first thing in the morning in a fasting state. If the blood was drawn immediately after the treadmill test for some reason, then you’d be right – something would need to be said about the effects on the O3I of being drawn in that non-resting state.
Line 80: There is no reference after "as described previously"?! Does OmegaQuant Analytics LLC use the same gas chromatographic method as described in reference 17? If so, it would be useful to mention it more clearly in the “Materials and Methods” section.
Yes, reference 17 is to the method used by OmegaQuant in this analysis. We have moved the [17] citation to immediately follow “previously” to remove any confusion on that point.
Lines 87-88: What is the rationale for excluding patients with prevalent CVD and BMI <18.5?
The BMI exclusion is the levels defined by the CDC as “below normal” and was used to avoid including men who may have had ongoing pathology and not yet known it. As regards excluding CVD, this is our traditional practice since we like to focus on a generally healthy cohort for our observational studies.
Discussion
The authors suggest that the difference observed between the study by Brasky et al. (2013) can be explained by the fact that the high levels of EPA and DHA in tissues and plasma may be an effect and not a cause of PCa due to increased delta-6-desaturase activity. It is therefore a pity that they did not answer this question by performing another analysis on the second visit.
What about industrial chemicals or pollutants found in high levels in farm raised fish, like PCBs, which were described to be associated with PCa (Ali I et al. Carcinogenesis, 2016, 37(12): 1144–1151)? Could this potential factor also affect the results of Brasky et al.?
This is a good point. We have now included the following in new Line 173ff: “Even if such speculation were true, and the slightly higher n-3 PUFA levels were caused by a higher intake of farmed salmon, it would be difficult to separate the effects of the n-3 PUFAs provided by the salmon from the effects of multiple environmental pollutants (Ali et al.) that may have been common in farmed salmon in those days (Foran et al.)”
The authors describe their population as being men of high social class and undoubtedly concerned by their health, of which 40.2% use omega-3 supplements. Maybe these people are not consuming too much farmed raised fish at affordable price but containing carcinogenic molecules, compared to other populations?!
We can’t really speculate on that. Actually, it would be likely that these men would eat more salmon, farmed or wild, than the average American man precisely because the former are more attuned to the importance of higher omega-3 intake, as evidenced by their high use of fish oil supplements.
Line 198: The parallel between self-reporting incident hypertension and PCa is not very relevant and seems to be only intended for self-citation.
We are sorry the reviewer saw it that way; that certainly was not our intention (i.e., to simply create an opportunity for self-citation). We were trying our best to address this potential limitation (self-reporting of adverse health outcomes), and it seemed most appropriate to do it using data from the Cooper population instead of some other unrelated cohort.
Reviewer 2 Report
The authors report the results of the association between n-3 PUFA and the risk of prostatic cancer (PCa). The results refer to a longitudinal study of 6274 patients taking n-3 PUFA and to a meta-analysis of the most relevant studies in this area. The authors conclude that they found no evidence between n-3 PUFA intake and the risk of PCa. In this context the longitudinal study shows several weaknesses as follows:
- the cause of prostate cancer is multifactorial and therefore it is difficult to establish how crucial the intake of n-3 PUFA can be;
- how long have patients been taking n-3 PUFA continuously? and at what dosage?
- why has testosteronemia not been evaluated in these patients? The correlation between testosteronemia and prostate cancer is now reported in the literature;
- the age range of the enrolled patients varied between 40 and 80 years: from 70 years onwards, patients are more likely to have PCa;
- The 5-year follow-up is too short to observe the onset of PCa: observation of at least 15 years should be necessary for patients enrolled at the age of 50.
Author Response
Reviewer 2
The authors report the results of the association between n-3 PUFA and the risk of prostatic cancer (PCa). The results refer to a longitudinal study of 6274 patients taking n-3 PUFA and to a meta-analysis of the most relevant studies in this area. The authors conclude that they found no evidence between n-3 PUFA intake and the risk of PCa. In this context the longitudinal study shows several weaknesses as follows:
- the cause of prostate cancer is multifactorial and therefore it is difficult to establish how crucial the intake of n-3 PUFA can be.
- No argument with that; it is difficult. This study was not trying to establish causation but to provide relevant data addressing earlier claims (unsubstantiated in our view, drawn from another observational study) that omega-3 intake “caused” PC.
- how long have patients been taking n-3 PUFA continuously? and at what dosage?
- We don’t have data on either of these points.
- Why has testosteronemia not been evaluated in these patients? The correlation between testosteronemia and prostate cancer is now reported in the literature.
- These patients were being seen for wellness exams as routinely done at Cooper Clinic. Testosterone measurements were not routinely included, hence we don’t have data on this.
- the age range of the enrolled patients varied between 40 and 80 years: from 70 years onwards, patients are more likely to have PCa;
- That’s true. Our subjects were in about the same age range as in the Brasky paper so it seemed reasonable to look here.
- The 5-year follow-up is too short to observe the onset of PCa: observation of at least 15 years should be necessary for patients enrolled at the age of 50.
- That would have been optimal, however, we only had data on the Omega-3 levels back 10 years and so our average follow-up of 5 years was all we had available to us. In response to some of these concerns we have added a couple more items to the Limitations section of the paper: “The five-year follow-up time was relatively short for men of average age 50, especially since PC is more likely to develop in older men.”